# Surgical Techniques and Outcomes of Colorectal Anastomosis after Left Hemicolectomy with Low Anterior Rectal Resection for Advanced Ovarian Cancer

**DOI:** 10.3390/cancers13164248

**Published:** 2021-08-23

**Authors:** Kyoko Nishikimi, Shinichi Tate, Ayumu Matsuoka, Satoyo Otsuka, Makio Shozu

**Affiliations:** Department of Gynecology, Chiba University Hospital, 1-8-1 Inohana, Chuo-ku, Chiba 260-8677, Japan; state@faculty.chiba-u.jp (S.T.); a-matsuoka@chiba-u.jp (A.M.); caxa5597@chiba-u.jp (S.O.); shozu@faculty.chiba-u.jp (M.S.)

**Keywords:** left hemicolectomy, low anterior resection, anastomosis, ovarian cancer, cytoreductive surgery

## Abstract

**Simple Summary:**

We tried to minimize the number anastomoses, restore intestinal continuity, and avoid stoma creation for 295 patients with stage III/IV ovarian cancer who underwent low anterior rectal resection (LAR) with or without colon resection during cytoreductive surgery. When remaining colon cannot reach the rectal stump after left hemicolectomy with LAR, we used the following three techniques for tension-free anastomosis: right colonic transposition, retro-ileal anastomosis through an ileal mesenteric defect, or additional colic artery division. The rate of stoma creation and rectal anastomotic was 3% (9/295) and 6.6% (19/286), respectively. Among 21 patients in whom the remaining colon did not reach the rectal stump after left hemicolectomy with LAR, 20 underwent tension-free anastomosis, including eight, six, and six patients undergoing right colonic transposition, retro-ileal anastomosis through an ileal mesenteric defect, and an additional colic artery division, respectively. Colorectal anastomosis is feasible in patients with extended colonic resection.

**Abstract:**

Extended colon resection is often performed in advanced ovarian cancer. Restoring intestinal continuity and avoiding stoma creation improve patients’ quality of life postoperatively. We tried to minimize the number of anastomoses, restore intestinal continuity, and avoid stoma creation for 295 patients with stage III/IV ovarian cancer who underwent low anterior rectal resection (LAR) with or without colon resection during cytoreductive surgery. When the remaining colon could not reach the rectal stump after left hemicolectomy with LAR, we used the following techniques for tension-free anastomosis: right colonic transposition, retro-ileal anastomosis through an ileal mesenteric defect, or an additional colic artery division. Rates of stoma creation and rectal anastomotic were 3% (9/295) and 6.6% (19/286), respectively. Among 21 patients in whom the remaining colon did not reach the rectal stump after left hemicolectomy with LAR, 20 underwent tension-free anastomosis, including eight, six, and six patients undergoing right colonic transposition, retro-ileal anastomosis through an ileal mesenteric defect, and an additional colic artery division, respectively. Colorectal anastomosis is feasible for patients with extended colonic resection. Low anastomotic leakage and stoma rates can be achieved with careful attention to colonic mobilization and tension-free anastomosis.

## 1. Introduction

No macroscopic residual disease after cytoreductive surgery is the most important prognostic factor for advanced ovarian cancer. Extended colonic resection is frequently necessary to achieve complete cytoreduction in patients with advanced ovarian cancer; in these cases, the rectum is the most commonly resected segment [1,2,3,4]. In addition to low anterior rectal resection, extended left or right hemicolectomy is performed in patients presenting with an omental cake that is densely adherent to the transverse colon with splenic or hepatic flexure. When extended colon resection is performed, protective ileostomy or colostomy is often performed [5] because there is a concern that anastomotic leakage will increase compared to the only rectal resection. However, complications involving the stoma or frequent bowel movements often occur, resulting in poor quality of life or poor compliance with continued adjuvant chemotherapy [6,7]. Therefore, gynecologists and patients would like to restore intestinal continuity and to avoid stoma creation if possible. Performing optimal surgical techniques for colorectal anastomosis after extended colon resection is necessary in patients who undergo cytoreductive surgery for advanced ovarian cancer.

Anastomosis between the remaining transverse or ascending colon and the rectal stump is considerably more difficult after left hemicolectomy with low anterior rectal resection than after low anterior rectal resection with or without right-sided colectomy (ileocecal resection or right hemicolectomy). This difficulty is because the remaining transverse or ascending colon frequently cannot reach the rectal stump in the pelvic cavity after left hemicolectomy with low anterior rectal resection. In patients with synchronous colorectal cancers or inflammatory bowel disease, right colonic transposition and retro-ileal anastomosis through an ileal mesenteric defect are used for tension-free anastomosis after extended left hemicolectomy [8,9,10,11,12,13]. Although several studies have reported the safety of these techniques [8,9,10,11,12,13], few reports have discussed their feasibility after left hemicolectomy with low anterior rectal resection in patients with advanced ovarian cancer.

We have tried to minimize the number anastomoses, to restore intestinal continuity, and to avoid stoma creation for patients with advanced ovarian cancer who underwent low anterior rectal resection with or without colon resection during cytoreductive surgery. We examined whether colorectal anastomosis was possible using right colonic transposition or retro-ileal anastomosis for advanced ovarian cancer when the remaining transverse or ascending colon could not reach the rectal stump after left hemicolectomy with low anterior rectal resection for advanced ovarian cancer. We investigated the rate of stoma creation and rectal anastomotic leakage in patients with advanced ovarian cancer who underwent low anterior rectal resection with or without extended colon resection when trying to restore intestinal continuity and to avoid stoma creation.

## 2. Materials and Methods

### 2.1. Patients

This retrospective study was approved by the Institutional Review Board of Chiba University Graduate School of Medicine (Number 3715). Between April 2008 and February 2021, 368 consecutive patients with International Federation of Gynecology and Obstetrics (FIGO) 2014 stage III/IV ovarian, fallopian tube, or primary peritoneal carcinoma underwent cytoreductive surgery for the primary treatment at Chiba University Hospital. Of these 368 patients, 313 underwent low anterior rectal resection with or without colon resection based on the site of dissemination. We excluded 18 patients who received long-term steroids for dermatomyositis or myasthenia gravis and those with septic shock or pre-septic shock secondary to cancer-induced rectal perforation because anastomosis was not planned due to the high risk of leakage. The 295 patients who underwent low anterior rectal resection with or without colon resection were included in this study. The definition of the colon resection was based on previous reports [14]: left hemicolectomy involved division of left colic artery and sigmoid artery, right hemicolectomy involved division of ileocolic artery and right colic artery and/or right branch of the middle colic artery, ileocecal resection involved division of ileocolic artery, and transverse colon involved division of middle colic artery.

### 2.2. Surgery and Chemotherapy

The surgical policy and chemotherapeutic regimen used during the study period were described in a previous report [15,16,17]. A monodisciplinary surgical team consisting of three gynecologic oncologists with expertise in advanced ovarian cancer treatment performed cytoreductive surgery, including colonic resection, to achieve complete cytoreduction. The surgical team selected the type of bowel resection and anastomosis that would minimize the number of anastomoses required while maintaining the maximum possible length of the healthy bowel during cytoreductive surgery. Ileostomy and colostomy were avoided as much as possible. Cytoreductive surgery was performed when complete cytoreduction could be achieved, regardless of primary debulking surgery or neoadjuvant chemotherapy, followed by interval debulking surgery. Even in stage IV patients, cytoreductive surgery was performed because residual tumor size after cytoreductive surgery was reported as a prognostic factor [18,19,20,21]. Paclitaxel and carboplatin with or without bevacizumab were administered as the first-line chemotherapy.

### 2.3. Surgical Techniques for Colorectal Anastomosis after Low Anterior Rectal Resection and Right-Sided Colectomy

Descending colon and splenic flexure were sufficiently mobilized to the lower border of the pancreas after low anterior rectal resection. If the descending colon did not reach the rectal stump, inferior mesenteric artery was divided. After confirming that the remaining colon reached the distal rectal stump without any tension, the end-to-end colorectal anastomosis was performed using the double staple technique.

### 2.4. Surgical Techniques for Tension-Free Colorectal Anastomosis after Left Hemicolectomy with Low Anterior Rectal Resection

The end-to-end colorectal anastomosis was performed using the double stapler technique after the remaining transverse or ascending colon reached the distal rectal stump. When the remaining transverse or ascending colon could not reach the distal rectal stump in the pelvic cavity after tumor resection, one of the following techniques were attempted to perform colorectal anastomosis, depending on the length and blood supply of the remaining colon. When the remaining colon did not reach the distal rectal stump even after trying three techniques, a permanent end stoma was created.

#### 2.4.1. Right Colonic Transposition

Right colonic transposition (Figure 1, Appendix A) was initiated with a complete mobilization of the right colon and hepatic flexure up to the base of the right mesocolon [8,9,10]. An incision was made along the Toldt’s fascia and was extended to the base of the right mesocolon and mesentery. The second and third duodenal segments were completely freed. The right-sided colon was transposed using one of the following procedures for tension-free anastomosis: (a) the right colon was rotated 180° in the sagittal plane and around the ileocecal pedicle axis. The bottom of the cecum was shifted to the level of the right upper abdomen, and the right colonic stump was shifted to the level of the rectal stump in the pelvis. (b) The right colon was rotated 180° in a counterclockwise manner in the frontal plane and around the superior mesenteric axis. The bottom of the cecum was shifted to the level of the mid or left upper abdomen, and the right colonic stump was shifted to the rectal stump in the pelvis.

#### 2.4.2. Retro-Ileal Anastomosis through an Ileal Mesenteric Defect

A window was created in an avascular area of the distal ileal mesentery (Figure 2). The transverse or ascending colon was passed through the window to reach the rectal stump [11,12,13].

#### 2.4.3. Additional Division of the Colic Artery

A branch of the middle colic artery was ligated and divided to improve the mobilization of the remaining transverse or ascending colon and enable the remaining colonic stump to reach the rectal stump without right colonic transposition or retro-ileal anastomosis (Appendix A).

### 2.5. Definition and Management of Anastomotic Leakage

Anastomotic leakage was diagnosed when the fecal fluid was identified from the pelvic drain placed near the anastomosis intra-operatively, or when CT revealed the fluid or air around the anastomosis and contrast agents injected from the drain or anus revealed the communication between the intra and extraluminal compartment at anastomosis.

Diverting colostomy or ileostomy was performed in patients who developed acute peritonitis. In patients who did not develop acute peritonitis, the drain was replaced and repositioned under radiographic guidance to ensure appropriate drainage. A new percutaneous drainage catheter was inserted in patients in whom the drain placed intra-operatively was not appropriately positioned. Antibiotics were administered to patients with signs of infection. Stop oral feeding with total parental nutrition were administered. After continuing drainage, drain was removed when fistulography revealed a simple tract without spread of the contrast agent in the pelvis surrounding the drain tract and anastomosis.

### 2.6. Surgical Outcomes

We investigated the rate of stoma creation and rectal anastomotic leakage, the surgical complexity scores [22], intraperitoneal residual tumor after cytoreductive surgery, postoperative complications according to the Clavien-Dindo classification [23], time interval until initiation of adjuvant chemotherapy, length of hospitalization, and survival.

### 2.7. Statistical Analysis

We used the chi-square or Wilcoxon non-parametric tests for comparisons of the clinical factors and anastomotic leakage. Survival rates were analyzed using the Kaplan–Meier method. All statistical analyses were performed using JMP software, version 11 (SAS, Cary, NC, USA). A *p*-value <0.05 was considered statistically significant.

## 3. Results

### 3.1. Patient Characteristics

Patient characteristics are summarized in Table 1. One hundred twenty-one patients (41.0%) have stage IV disease. Thirty-three patients were diagnosed with stage IVB due to cardiophrenic lymph node metastasis. Seventeen of 24 patients with stage IVA had positive washing cytology of pleural effusion during interval debulking surgery after neoadjuvant chemotherapy. The median peritoneal cancer index [24,25] was 16. Neoadjuvant chemotherapy followed by interval debulking surgery was performed in 178 patients (60.3%). Forty patients underwent left hemicolectomy with low anterior rectal resection (15 who underwent concomitant ileocecal resection, and 5 who underwent concomitant right hemicolectomy), 105 patients underwent right-sided colectomy (69 who underwent ileocecal and concomitant low anterior rectal resection, and 36 who underwent right hemicolectomy and concomitant low anterior rectal resection), and 145 patients underwent only low anterior rectal resection. One patient underwent total colectomy, and four patients underwent low anterior rectal resection concomitant with transverse colectomy.

### 3.2. Surgical Outcomes

Surgical outomes are summarized in Table 2. Permanent colostomy was performed in nine (6.6%) patients. The reasons were as follows: a residual distal rectum too short to perform anastomosis because of the excision line on the anal side of the rectum being near the levator ani muscle in seven patients; the remaining colon not reaching the distal rectal stump even after applying several techniques for mobilization in one patient; and blood supply in the distal part of the residual colon after LAR being insufficient in one patient. Protective ileostomy was performed in four (1.4%) patients. The reasons were long operative time with massive bleeding in three patients, and cecum-rectal anastomosis in one patient.

Rectal anastomotic leakage occurred in 19 of 295 patients (3 of 40 who underwent left hemicolectomy with low anterior rectal resection, 9 of 105 who underwent right-sided colectomy with low anterior rectal resection, and 7 of 145 who underwent only low anterior rectal resection). The median onset of anastomotic leakage was 9 days after cytoreductive surgery. Fifteen patients were successfully treated with percutaneous drainage without re-operation. The median duration of drainage was 26 days. Four patients required re-operation to create an ileostomy (one patient developed acute peritonitis, and three patients did not cure by continuous percutaneous drainage).

Clavien–Dindo Grade IIIb postoperative complication within 30 days of surgery occurred in 8 patients: rectal anastomotic leakage in 4, ureteral leakage in 2, vesico-vaginal-rectal fistula in 1, pancreatic fistula in 1. Grade IV of postoperative bleeding and small bowel syndrome occurred in one patient. Grade V of postoperative bleeding and multiple organ failure occurred in one patient.

### 3.3. Rate and Techniques of Colorectal Anastomosis after Left Hemicolectomy with Low Anterior Rectal Resection

Of the 40 patients who underwent left hemicolectomy with low anterior rectal resection, 39 (97%) underwent tension-free colorectal anastomosis as follows: 8 underwent right colonic transposition, 6 underwent retro-ileal anastomosis through an ileal mesenteric defect, 6 underwent only additional colic artery division, and 19 underwent no extra techniques. The colic artery was divided over multiple segments along its course. The median length that was continuously resected from the rectum to the transverse or ascending colon was 79 (43–100) cm, 47 (31–65) cm, and 27 (23–55) cm in patients who underwent right colonic transposition, retro-ileal anastomosis, and additional colic artery division, respectively.

Among the 40 patients who underwent left hemicolectomy with low anterior rectal resection, in one patient, the remaining transverse colon did not reach the distal rectal stump after right and left hemicolectomy with low anterior rectal resection even after the aforementioned additional techniques were attempted; this patient did not undergo anastomosis and a permanent transverse colostomy was performed.

No patient underwent protective ileostomy, among the 39 patients who underwent left hemicolectomy with primary colorectal anastomosis.

### 3.4. Clinical Factors for Rectal Anastomotic Leakage

Patients with and without rectal anastomotic leakage did not differ in the age, stage, serum albumin level before treatment, peritoneal cancer index, surgical complexity score, residual tumor, type of large bowel resection, length that was continuously resected from the rectum to the colon, or number of intestinal anastomoses (Table 3). The performance status before the treatment was poorer in patients who developed rectal anastomotic leakage than in those who did not (*p* = 0.02). Anastomotic leakage occurred more frequently in patients who underwent interval debulking surgery after neoadjuvant chemotherapy than in those who underwent primary debulking surgery (*p* = 0.03).

### 3.5. Survival

The median follow-up period was 40 months. In patients with and without anastomotic leakage, the median durations to initiation of adjuvant chemotherapy were 51 and 27 days, respectively, and lengths of hospital stay were 52 and 28 days, respectively. However, their prognoses did not differ. Median progression-free survivals of patients with and without anastomotic leakage were 33 and 34 months, respectively (*p* = 0.93, log-rank). Median overall survivals of patients with and without anastomotic leakage were not reached and 101 months, respectively (*p* = 0.76, log-rank) (Figure 3).

## 4. Discussion

Our study showed low rates of stoma creation and rectal anastomotic leakage in patients with advanced ovarian cancer who underwent low anterior rectal resection with or without extended colon resection when trying to restore intestinal continuity and avoid stoma creation. Anastomotic leakage is expected to occur more frequently in complicated colorectal surgery than in simple colorectal surgery. However, low anastomotic leakage and stoma rates were achieved with careful attention to colonic mobilization and tension-free anastomosis in this study. Although anastomosis between the remaining transverse or ascending colon and the rectal stump is considerably more difficult after left hemicolectomy with low anterior rectal resection than after other types of colon resection, tension-free anastomosis can be performed using one of the following procedures: right colonic transposition, retro-ileal anastomosis via an ileal mesenteric defect, and an additional colic artery division.

The choice of the anastomotic technique was determined based on the site, length, and blood supply of the remaining colon. Right colonic transposition was feasible in patients without disseminated cancer in the ileocecal area and with a long length of the colon that was continuously resected from the rectum to the transverse or ascending colon. Middle colic artery division was necessary in such cases for mobilization of the residual right colon. Retro-ileal anastomosis through an ileal mesenteric defect was feasible in patients who had disseminated cancer in the ileocecal area. The division of the left branch of the middle colic and/or inferior mesenteric artery was used to achieve tension-free anastomosis in patients with a relatively long residual transverse colonic segment.

The remaining transverse or ascending colon is less likely to reach the distal rectal stump after left hemicolectomy with low anterior rectal resection than after low anterior rectal resection with or without right hemicolectomy because the transverse mesocolon and the middle colic artery are short. Saunders et al. reported that, among those who underwent laparotomy, the mid-transverse colon did not reach the symphysis pubis or lower in 71% (84/118) of patients [26]. Therefore, middle colic artery division provides additional length of residual transverse colon and facilitates its mobilization. Following the division of the left branch of the middle colic artery, the remaining colonic stump can usually reach the rectal stump with or without retro-ileal anastomosis. Right colonic transposition can be performed following the division of the right branch of the middle colic artery for better mobilization of the mesentery of the right colon. Careful preservation of the marginal arcade and its collaterals ensures the maintenance of blood supply even after the division of the middle colic artery.

Left hemicolectomy with low anterior rectal resection was not associated with any severe complications in this study. The rate of anastomotic leakage after left hemicolectomy with low anterior rectal resection was not higher than that after low anterior rectal resection with or without right-sided colectomy. The aforementioned surgical techniques for anastomosis contributed to the low incidence of anastomotic leakage. Right colonic transposition has a risk of torsion of the residual colon, while retro-ileal anastomosis has a risk of internal hernia. However, these complications did not occur in our study. Extensive intraperitoneal surgery may result in postoperative adhesions, which prevent torsion or internal hernia.

Protective ileostomy was not performed in any patient who underwent left hemicolectomy with low anterior rectal resection in our study. Anastomotic leakage occurred in three patients (7.5%) Left hemicolectomy with low anterior rectal resection; however, all patients were successfully treated with percutaneous drainage. Silver et al. reported that protective ileostomy was performed in 18 of 19 patients with ovarian cancer who underwent extended left hemicolectomy for adjuvant intraperitoneal chemotherapy [27]. Although it is difficult to directly compare the rates of anastomotic leakage and protective ileostomy between our study and previous studies, our results suggest that protective ileostomy is not always necessary after left hemicolectomy with low anterior rectal resection.

Compared with right-sided colectomy, left-sided colectomy is less commonly performed for ovarian cancer. Except for the findings reported by Silver et al. (21%), the left hemicolectomy rates were <6% in patients who underwent cytoreductive surgery [1,27,28,29,30,31,32]. We could not directly compare the rates of left hemicolectomy observed in our study (14%) with those of previous studies because of the varied patient characteristics between the study groups. Most studies have included only patients who underwent primary debulking surgery and consequently achieved residual disease <1 cm. Patients who need to undergo left hemicolectomy during primary debulking surgery show widespread dissemination of the intra-abdominal tumor in addition to omental cake invasion of the splenic flexure. In such cases, left hemicolectomy may not be useful for residual tumors other than splenic flexure lesions, and widespread dissemination is likely to result in residual disease, even in those who undergo left hemicolectomy. In contrast, 70% of the patients who underwent left hemicolectomy in this study underwent interval debulking surgery. For patients in whom the tumor volume in the abdomen was reduced by neoadjuvant chemotherapy, left hemicolectomy was a useful surgery to achieve no residual disease. Several studies have shown that an increasing number of patients undergo neoadjuvant chemotherapy followed by interval debulking surgery, and phase III trials have proved that this therapeutic strategy is not inferior to primary debulking surgery with regard to survival rates [33,34,35,36]. Therefore, it is reasonable to conclude that indications for left hemicolectomy as a treatment for advanced ovarian cancer are likely to increase in the future.

The following are the limitations of our study: (a) We did not compare the prognosis between patients who underwent colostomy and those who underwent colorectal anastomosis. It is unclear whether patients who undergo colorectal anastomosis show a better prognosis than those who undergo protective ileostomy or permanent colostomy. (b) We did not investigate the long-term quality of life. (c) The retrospective design of this small-scale study is a drawback; however, this study is among the largest studies performed in patients who underwent left hemicolectomy for advanced ovarian cancer.

## 5. Conclusions

Colorectal anastomosis is eminently feasible in patients with extended colonic resection. Low anastomotic leakage and stoma rates can be achieved with careful attention to colonic mobilization and tension-free anastomosis. Colorectal anastomosis after left hemicolectomy with low anterior rectal resection were feasible using right colonic transposition, retro-ileal anastomosis through an ileal mesenteric defect, or additional colic artery division. Further prospective studies are warranted to investigate the long-term quality of life in patients with advanced ovarian cancer who undergo low anterior rectal resection with extended colon resection.

## Figures and Tables

**Figure 1 cancers-13-04248-f001:**
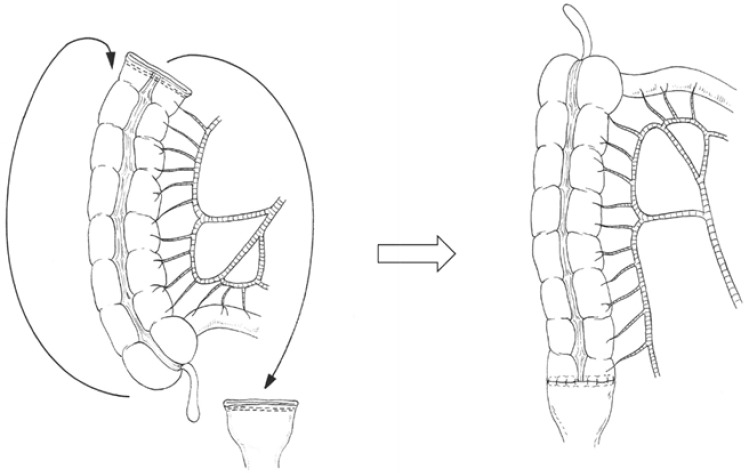
Right colonic transposition for colorectal anastomotic techniques after left hemicolectomy with rectal low anterior resection. The right colon is rotated 180° in the sagittal plane and around the ileocecal pedicle axis or the right colon is rotated 180° in a counterclockwise manner in the frontal plane and around the superior mesenteric vessel axis.

**Figure 2 cancers-13-04248-f002:**
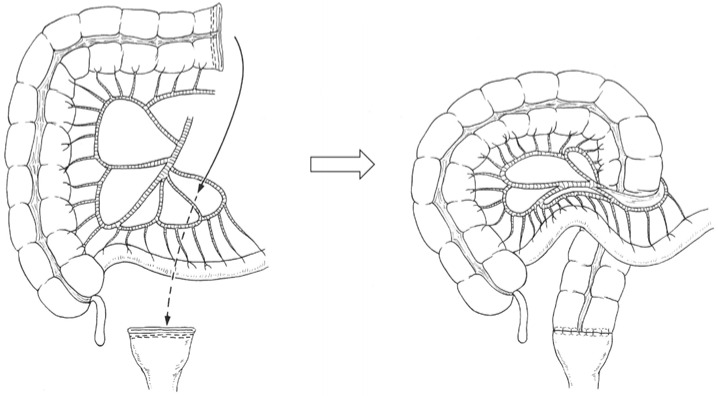
Retro-ileal anastomosis through an ileal mesenteric defect for colorectal anastomotic techniques after left hemicolectomy with rectal low anterior resection. The transverse or right colon is passed through a window created in an avascular area of the distal ileal mesentery to reach the rectal stump.

**Figure 3 cancers-13-04248-f003:**
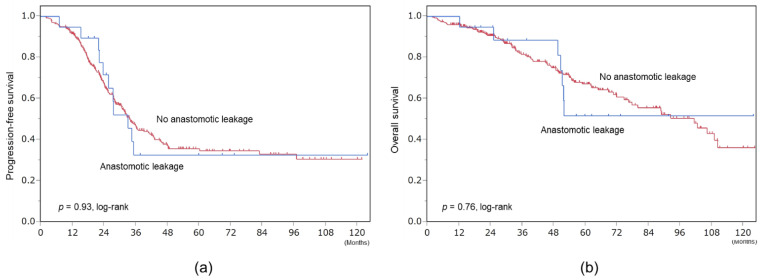
Survival. (**a**) Progression-free survival. (**b**) Overall survival.

**Table 1 cancers-13-04248-t001:** Patients characteristics.

Variables	All	Left Hemicolectomy with Low Anterior Rectal Resection	Right-Sided Colectomy with Low Anterior Rectal Resection	Transverse Colectomy with Low Anterior Rectal Resection or Total Colectomy	Low Anterior Rectal Resection Only
	*n* = 295	*n* = 40	*n* = 105	*n* = 5	*n* = 145
Age, median (IQR)	63 (51–71)	65 (51–71)	66 (54–72)	62 (61–70)	59 (50–69)
Primary site					
Ovary	175 (59.3%)	23 (57.5%)	51 (48.6%)	5 (100%)	96 (66.2%)
Fallopian tube	102 (34.6%)	15 (37.5%)	46 (43.8%)	0 (0%)	41 (28.3%)
Peritoneum	18 (6.1%)	2 (5.0%)	8 (7.6%)	0 (0%)	8 (5.5%)
Performance status					
0	69 (23.3%)	8 (20.0%)	20 (19.1%)	0 (0%)	41 (28.3%)
1	135 (45.8%)	24 (60.0%)	43 (41.0%)	2 (40.0%)	66 (45.5%)
2	66 (22.4%)	8 (20.0%)	31(30.0%)	2 (40.0%)	25(17.2%)
3	25(8.5%)	0(0%)	11(10.5%)	1 (20.0%)	13(9.0%)
FIGO 2014 stage					
IVA	28 (9.5%)	0 (0%)	1 (1.0%)	0 (0%)	27 (18.6%)
IVB	15 (5.1%)	0 (0%)	1 (1.0%)	0 (0%)	14 (9.7%)
IVC	131 (44.4%)	18 (45.0%)	55 (52.4%)	4 (80.0%)	54 (37.2%)
IIIA	24 (8.1%)	3 (7.5%)	11 (10.5%)	0 (0%)	10(6.9%)
IIIB	97 (32.9%)	19 (47.5%)	37 (35.2%)	1 (20.0%)	40 (27.6%)
Histology					
High-grade serous	223 (75.6%)	31 (77.5%)	90 (85.7%)	3 (60.0%)	99 (68.3%)
Non high-grade serous	72 (24.4%)	9 (22.5%)	12 (14.3%)	2 (40.0%)	45 (31.7%)
Clear	25 (8.5%)	5 (12.5%)	5 (4.8%)	0 (0%)	15 (10.3%)
Endometrioid	18 (6.1%)	1 (2.5%)	4 (3.8%)	0 (0%)	13 (8.9%)
Mucinous	1 (0.3%)	1 (2.5%)	0 (0%)	0 (0%)	0 (0%)
Others	28 (9.5%)	2 (5.0%)	6 (5.7%)	2 (40.0%)	18 (12.4%)
Timing of cytoreductive surgery					
Primary	117 (39.7%)	13 (32.5%)	22 (21.0%)	1 (20.0%)	81 (55.9%)
Interval	178 (60.3%)	27 (67.5%)	83 (79.1%)	4 (80.0%)	64 (44.1%)
Peritoneal cancer index, median (IQR)	16 (8–21)	23 (18–26)	20 (16–23)	20 (10–23)	9 (5–16)

Abbreviations: FIGO: The International Federation of Gynecology and Obstetrics, IQR: interquartile range.

**Table 2 cancers-13-04248-t002:** Surgical outcome.

Variables	All	Left Hemicolectomy with Low Anterior Rectal Resection	Right-Sided Colectomy with Low Anterior Rectal Resection	Transverse Colectomy with Low Anterior Rectal Resection or Total Colectomy	Low Anterior Rectal Resection Only
	*n* = 295	*n* = 40	*n* = 105	*n* = 5	*n* = 145
Permanent colostomy	9 (3.1%)	1 (0.3%)	2 (2.0%)	0(0%)	6 (4.1%)
Protective ileostomy	4 (1.4%)	0 (0%)	1 (0.1%)	1 (20.0%)	2 (1.4%)
Rectal anastomotic leakage	19 (6.4%)	3 (7.5%)	9 (8.6%)	0 (0%)	7 (4.8%)
Conservative treatment with percutaneous drainage	15 (5.1%)	3 (7.5%)	9 (8.6%)	0 (0%)	3 (2.1%)
Requiring re-operation	4 (1.4%)	0 (0%)	0 (0%)	0 (0%)	4 (2.8%)
Surgical complexity score, median (IQR)	13 [10,11,12,13,14,15]	15 (13–16)	15 (14–16)	11 (9–16)	11 (7–13)
Residual tumor					
0	270 (91.5%)	33 (82.5%)	97 (92.4%)	3 (60.0%)	137 (94.5%)
0.1–1.0 cm	21 (7.1%)	5 (12.5%)	7 (6.7%)	2 (40.0%)	7 (4.8%)
>1.0 cm	4 (1.4%)	2 (5.0%)	1 (1.0%)	0 (0%)	1 (0.7%)
Postoperative complications					
Clavien-Dindo IIIa	61 (20.7%)	15 (37.5%)	30 (28.6%)	0 (0%)	16 (11.0%)
Clavien-Dindo IIIb-V	11 (3.7%)	1 (2.5%)	1 (1.0%)	0 (0%)	9 (6.2%)
Time to initiation of adjuvant chemotherapy, median, day	27 [21,22,23,24,25,26,27,28,29,30,31,32,33,34]	27 (21–38)	27 (22–35)	29 (28–37)	26 (20–32)
Length of hospitalization, median, day	29 (23–40)	28 (23–47)	29 (23–38)	29 (22–40)	49 (21–51)

Abbreviations: IQR: interquartile range.

**Table 3 cancers-13-04248-t003:** Clinical factors for rectal anastomotic leakage.

Variables	Rectal Anastomotic Leakage (+)	Rectal Anastomotic Leakage (−)	*p* Value
	*n* = 19	*n* = 276	
Age, median (IQR)	62 (51–70)	63 (51–71)	0.86
Performance status			0.02
0–1	8 (42.1%)	196 (71.0%)	
2–3	11 (57.9%)	80 (29.0%)	
FIGO 2014 stage			0.92
III	11 (57.9%)	163 (57.9%)	
IV	8 (42.1%)	113 (42.1%)	
Serum albumin before cytoreductive surgery (g/dL), median (IQR)	3.9 (3.7–4.1)	3.9 (3.5–4.2)	0.98
Timing of cytoreductive surgery			0.03
Primary	3 (15.8%)	114 (41.3%)	
Interval	16 (84.2%)	162 (58.7%)	
Peritoneal cancer index, median (IQR)	16 (8–21)	18 (16–25)	0.06
Surgical complexity score, median (IQR)	13 (13–15)	13 (10–15)	0.16
Residual tumor			0.60
Microscopic	18 (94.7%)	252 (91.3%)	
Macroscopic	1 (5.3%)	24 (8.7%)	
Type of large bowel resection			0.61
Left hemicolectomy with low anterior rectal resection	3 (15.8%)	37 (13.4%)	
Right-sided colectomy with low anterior rectal resection	9 (47.4%)	96 (34.8%)	
Transverse colectomy with low anterior rectal resection or total colectomy	0 (0%)	5 (1.8%)	
Low anterior rectal resection only	7 (36.8%)	138 (50.0%)	
Length that was continuously resected from the rectum to the colon (cm), median (IQR)	21 (16–30)	22 (17–29)	0.79
Number of intestinal anastomosis			0.19
0 or 1	7	159	
2 or 3	11	117	

Abbreviations: FIGO: The International Federation of Gynecology and Obstetrics, IQR: interquartile range.

## Data Availability

All datasets analyzed during the current study are available from the corresponding author upon reasonable request.

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
