# Peer review of "Surgical Techniques and Outcomes of Colorectal Anastomosis after Left Hemicolectomy with Low Anterior Rectal Resection for Advanced Ovarian Cancer"

_cancers, 2021, doi:10.3390/cancers13164248_

Round 1

Reviewer 1 Report

The end point of this study is not well centered. Complications in cytoreductive surgery are mainly due to other factors: PCI, involvement of the small intestine, previous chemotherapies, etc. Other factors not directly related to the type of anastomosis or transposition influences the incidence rate of dehiscence of colo-rectal and ileocolic anastomoses.

Reasons for leakage and dehiscence are identifiable in: vascularization of the stumps, local infections or increased pressure in the bowel from prolonged intestinal paralysis and anastomotic traction.In cytoreduction for any type of carcinosis the colonic anastomosis is only one of the complication of the procedure also in general surgery.Morever, it is necessary to highlight how cytoreduction is indicated in ovarian neoplasms Figo 3c and not in Figo 4 as a cytoreductive surgery has a value on DFS only in case of locoregional disease cc0 and not in metastatic diseases. Prognosis of patient does not rely on anastomotic technique but on disease extension ! 

Reviewer 2 Report

The described procedures are commonly performed during abdominal extensive cytoreduction. Right colon transposition as well as single branch colic artery division are good options in selected cases, whereas retro-ileal anastomosis is generally considered more risky and as a consequence little used  in the surgical community.

Having said that, the true problem of any colorectal anastomosis is the leakage. It has a multifactorial origin and the surgical technique is only one of the component. Overall incidence rates are within the reported ranges in the literature, but several aspects need to be clarified. First, an accurate definition of leakage is lacking (clinical or radiological ?). Second, how the diagnosis of leakage was established ?  Third, the timing of onset is another fundamental aspect (early or late ?).  Fourth, more details about conservative treatment are needed, since percutaneous drainage is a too generic term and which was was the indication for a stoma ? Fifth, besides time to chemotherapy, the length of hospital stay was higher patients with leakages  ? ( eg, how much time was required for recovering). Sixth, a detailed description of complication types and deaths should be included. Prognosis basically depends on the tumours extension and not anastomotic type, so the surival calculation is unnecessary for the purpose of the study. Last but not least, a surprisingly high incidence of stage IV (especially IV b) was included, but no residual tumour was found in the great majority of patients: how could you explain this discrepancy ? Did you take into account the option of HIPEC for these patients ?

Reference number 15 is incomplete

Reviewer 3 Report

This paper describes colonic resection and anastomotic techniques after surgery for advanced ovarian cancer. As outcomes they report stoma rates, anastomotic leak rates and survival curves (disease free and overall survival). The authors appear to have an aggressive cytoreductive approach to the ovarian cancer which requires extended colonic resection if there is significant omental disease. With regard to colonic resection their approach is to minimise the number anastomoses required and are aiming to restore intestinal continuity and avoid a stoma in all cases.

They attempt to compare outcomes for 3 groups which they have defined as low anterior resection with left hemicolectomy, right sided colectomy with low anterior resection and low anterior resection only. Their hypothesis being that a colorectal anastomosis in pateints with left hemicolectomy and low anterior resection will be technically more challenging than in the other groups which might show as differences in leak and stoma rates.

They report no significant difference in their three groups for stoma rates and leak rates. The stoma rate and leak rate for the left hemicolectomy and low anterior resection group was 0.3% and 7.5% respectively.

Comments for consideration:

I was unable to view the videos. I am am sure these would be of interest to the target audience

The authors are to be congratulated on their low stoma rate and leak rates.  Leak rates for "uncomplicated" anterior resection would be in the region of 6%. One would expect the complex surgery decribed in this paper to have a higher leak rate. I also feel that the authors low stoma rate is exceptional I suspect many surgeons would have a much higher stoma rate. So publishing this paper would highlight the surgical possiblities to the wider audience.

Methods: The authors state 295 patients had a low anterior resection these are subdivided into groups: 40 low anterior resection with left hemi colectomy, 105 Low anterior resection and right sided resection and finally 145 low anterior resection alone. 5 patients are excluded because of small numbers in each catagory.

This division into groups is, to me, arbitary and contrived particularly when one considers the resaons for the 5 exclusions. One senses the authors have done this simply to generate some comparative statistics. These are of little use since they are only comparing patients on their own series and not in other centres. The left hemicolcetomy group is small and very herterogenous. The authors do not define what a left hemicolectomy is in their series and the attempts to divide the patients into groups leads to clumsy descriptions which makes the text confusing particularly when trying to interpret the results. I found table 2 confusing and difficult to understand. The addition of the term "Extended Left Hemicolectomy" line 284 only serves to confuse further. Perhaps a diagram showing the proximal resection line and where the arteries have been divided for each group would help with understanding.

To me the simple message of the paper is: colorectal anastomosis is eminently feasible in patients with extended colonic resection. Low leak and stoma rates can be achieved with careful attention to colonic mobilisation. In the light of my comments above I wonder if the authors should simply present these results for the 295 patients in their series rather than artificially divide them into the groups mentioned above. This would give a headline stoma rate of 9/295 (3%) and a leak rate of 19/286 (6.6%). These are very respectable figures and a much clearer message.

Round 2

Reviewer 1 Report

no

Reviewer 2 Report

None